# Post-translational modifications of *Drosophila melanogaster* HOX protein, Sex combs reduced

**Anirban Banerjee**\*, **Anthony Percival-Smith**

Department of Biology, The University of Western Ontario, London, Ontario, Canada

\* abanerj7@uwo.ca

## Abstract

Homeotic selector (HOX) transcription factors (TFs) regulate gene expression that determines the identity of *Drosophila* segments along the anterior-posterior (A-P) axis. The current challenge with HOX proteins is understanding how they achieve their functional specificity while sharing a highly conserved homeodomain (HD) that recognize the same DNA binding sites. One mechanism proposed to regulate HOX activity is differential post-translational modification (PTM). As a first step in investigating this hypothesis, the sites of PTM on a Sex combs reduced protein fused to a triple tag (SCRTT) extracted from developing embryos were identified by Tandem Mass Spectrometry (MS/MS). The PTMs identified include phosphorylation at S185, S201, T315, S316, T317 and T324, acetylation at K218, S223, S227, K309, K434 and K439, formylation at K218, K309, K325, K341, K369, K434 and K439, methylation at S19, S166, K168 and T364, carboxylation at D108, K298, W307, K309, E323, K325 and K369, and hydroxylation at P22, Y87, P107, D108, D111, P269, P306, R310, N321, K325, Y334, R366, P392 and Y398. Of the 44 modifications, 18 map to functionally important regions of SCR. Besides a highly conserved DNA-binding HD, HOX proteins also have functionally important, evolutionarily conserved small motifs, which may be Short Linear Motifs (SLiMs). SLiMs are proposed to be preferential sites of phosphorylation. Although 6 of 7 phosphosites map to regions of predicted SLiMs, we find no support for the hypothesis that the individual S, T and Y residues of predicted SLiMs are phosphorylated more frequently than S, T and Y residues outside of predicted SLiMs.

## Introduction

The identity of body segments along the Anterior-Posterior (A-P) axis of Bilaterians is determined by a set of developmental control genes called *Homeotic selector* (*Hox*) genes [1,2]. These genes encode transcription factors (TFs) that regulate expression of target genes by binding to DNA-binding sites with a 60 amino acid DNA-binding homeodomain (HD) [3]. That HOX proteins determine distinct segmental identities and regulate distinct patterns of gene expression while recognizing similar DNA binding sites is a major paradox. Interactions with the cofactor Extradenticle (EXD) is one mechanism proposed to mediate the functional

**Funding:** This work was supported by the Natural Sciences and Engineering Research Council of

Canada (NSERC) discovery grant No. RGPIN-2015-06356 to AP-S (URL: http://www.nserc-crsng.gc.ca/index_eng.asp). The funders had no role in study design, data collection and analysis, decision to publish, or preparation of the manuscript.

**Competing interests:** The authors have declared that no competing interests exist.

specificity of HOX proteins. In addition to HOX cofactor protein interactions, PTMs are also proposed to have a role in the regulation of functional specificity [4–6]. Here the PTMs of a HOX protein, Sex combs reduced (SCR) are mapped as a first step towards understanding this regulation of HOX protein activity.

SCR is a well-studied HOX protein which is essential for the formation of larval salivary glands, adult proboscis and adult prothoracic legs in *Drosophila melanogaster* [7–11]. Besides the highly conserved HD, which is essential for function, there are other small conserved motifs in SCR, the N-terminal octapeptide motif (MSSYQFVNS), LASCY motif, DYTQL motif, NEAGS motif, YPWM motif, NANGE motif, KMAS motif and the C-terminal domain (CTD), that are also important for function (Fig 1A) [6,11–16]. Amino acid substitutions in these small conserved motifs affect SCR activity both for proboscis and sex comb determination and for the suppression of ectopic proboscis formation [11,14,15]. The non-uniform effect of *Hox* mutant alleles across tissues, termed differential pleiotropy, suggest that the conserved

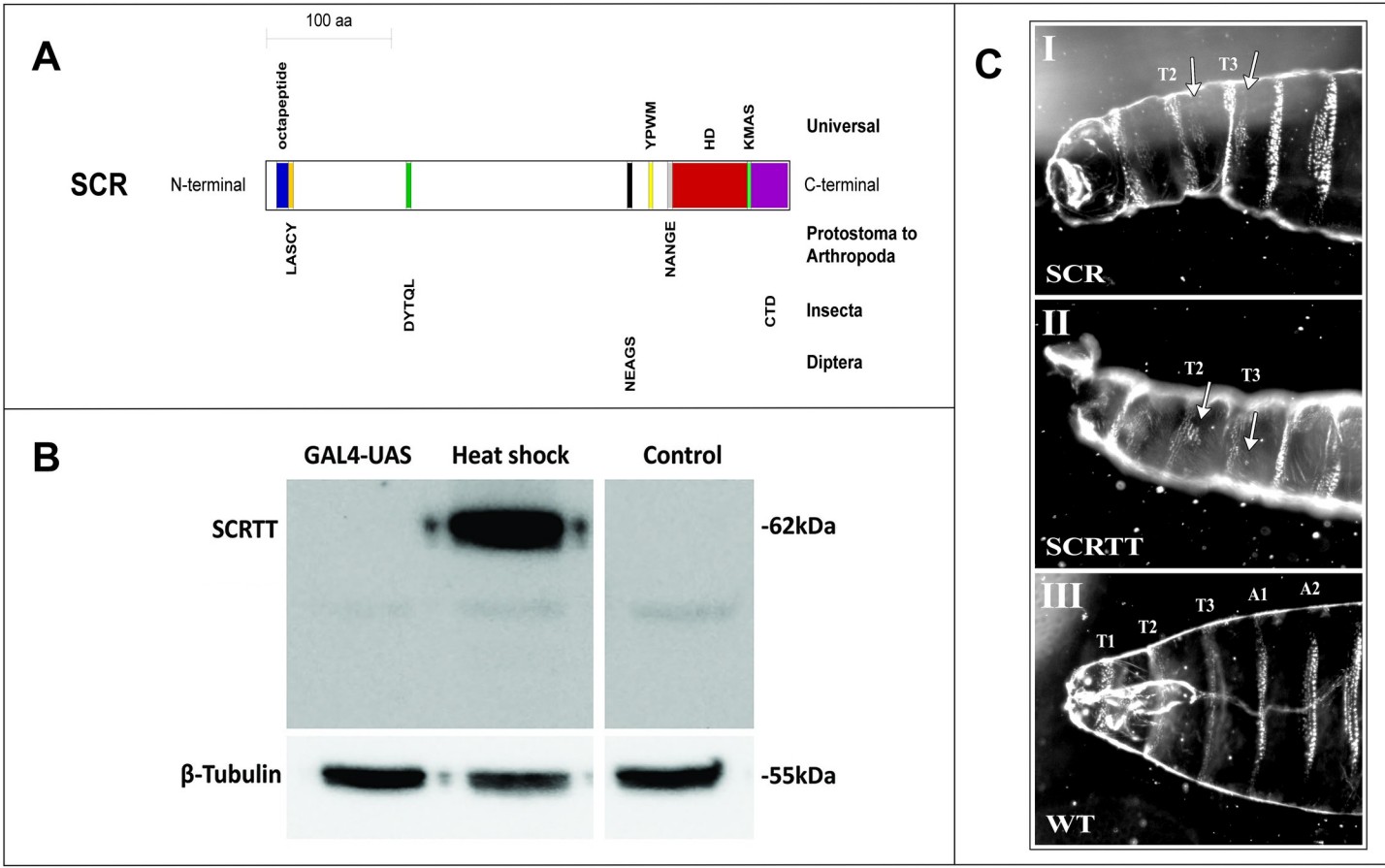

**Fig 1. Sex combs reduced functional domains/motifs, expression and phenotype.** (A) Schematic of SCR protein showing highly conserved, functional domains/motifs. The block diagram is drawn to scale and the domains/motifs are color-coded. The octapeptide motif is labeled in blue, LASCY motif in orange, DYTQL motif in dark green, NEAGS motif in black, YPWM motif in yellow, NANGE motif in grey, HD in dark red, KMAS motif in light green and CTD in purple. The taxonomic level of conservation of the domains/motifs is indicated on the right. (B) Comparison of the expression of SCRTT protein from heat-shock and *UAS* promoters. The method of expression is indicated above the lanes of the Western Blot of protein extracted from embryos. The position of SCRTT is indicated on the left, and the positions of molecular weight markers are indicated on the right. The level of β-tubulin expression is shown at the bottom. *y w* is the untransformed control. (C) First instar larval cuticular phenotypes of embryonic ectopic expression of SCR and SCRTT proteins. The anterior half of the larva is shown. The untagged SCR protein expressed with the GAL4-*UAS* system using a ubiquitous *armadillo*-GAL4 is shown in panel (I); whereas, the SCRTT protein expressed from a heat-shock promoter is shown in panel (II). T1, T2 and T3 refer to first, second and third thoracic segments. A1 and A2 refer to first and second abdominal segments. (I) and (II) Ectopic expression of SCR and SCRTT, respectively (T2 and T3 beards marked with arrows). (III) Control wild-type (WT) first instar larval cuticle.

domains and motifs contribute to HOX activity in a small, additive and tissue-specific manner [6,11,14,15,17–19]. The short, conserved motifs found in SCR and other HOX proteins may also be Short Linear Motifs (SLiMs) [6]. SLiMs or Eukaryotic Linear Motifs (ELMs) are short stretches of protein sequence (typically 3–10 amino acids long containing 2–3 specificity-determining residues) present in regions of intrinsic disorder [20–23]. SLiMs function as sites of protein-protein interaction, PTMs and cell compartment targeting signals [20–28]. Experimentally validated SLiMs have been curated into the ELM database and the database is used to predict SLiMs [28]. Although SLiMs have a low information content and are predicted to occur frequently in the proteome, the 22% of human disease mutations that occur in intrinsically disordered regions of human proteins map frequently to functionally important residues of validated SLiMs and also map frequently to predicted SLiMs suggesting that predicted SLiM sequences are correlated with function [29]. Viral proteins have SLiMs that mimic the SLiMs in host proteins which enables the virus to evade host immune functions [30,31]. Phosphorylation of the PDZ-binding motif (PBM), which is a SLiM, in the disordered C-terminal tail of ribosomal S6 kinase 1 (RSK1) protein is important for epidermal growth factor regulation [32]. The annotation of SLiMs in the ELM database suggest that a significant proportion are targets of phosphorylation.

Phosphorylation is a major regulatory mechanism used in cellular signaling pathways that often terminate in the regulation of TF activity and subsequent regulation of gene expression [33–35]. HOX are phosphoproteins [36–47]. PP2A-B' is proposed to activate SCR by dephosphorylating residues in the N-terminal arm of the SCR HD that when phosphorylated inhibit interaction of the SCR HD with DNA. Indeed, peptides with N-terminal arm sequence are phosphorylated by cAMP-dependent protein kinase A (PKA) and dephosphorylated by Serine-threonine protein phosphatase 2A (PP2A-B') *in vitro* [36]. However, a null *PP2A-B'* allele does not affect SCR activity suggesting that dephosphorylation by PP2A-B' plays no role in regulating SCR activity [48]. A drawback of the methodologies employed to study phosphorylation of SCR and other *Drosophila* HOX and HD-containing proteins so far is they provide no direct and definitive information on which amino acid residue is phosphorylated in the developing *Drosophila* embryo [36–47]. We have used Tandem Mass Spectrometry (MS/MS) on SCR extracted from developing embryos to provide data about a possible repertoire of PTMs [49].

## Materials and methods

### *Drosophila* husbandry

The fly stocks were maintained at 23˚C and 60% humidity. The flies were grown in 20ml vials and ~300ml bottles containing corn meal food (1% (w/v) *Drosophila*-grade agar, 6% (w/v) sucrose, 10% (w/v) fine-ground cornmeal, 1.5% (w/v) yeast and 0.375% (w/v) 2-methyl hydroxybenzoate–an anti-fungal agent). To collect embryos, flies were allowed to lay eggs/embryos on apple juice plates (2.5% (w/v) *Drosophila*-grade agar, 6% (w/v) sucrose, 50% apple juice and 0.3% (w/v) 2-methyl hydroxybenzoate) smeared with a yeast paste (with 0.375% (w/v) 2-methyl hydroxybenzoate).

### Generation of *ScrTT* fusion constructs

A *NotI* DNA fragment encoding SCR fused to the triple tag (TT) (containing 3X FLAG, Strep II and 6X His tags) [50] was isolated from a pUAST construct [11] and inserted into pET-3a and pCaSpeR [51–53] using standard molecular cloning techniques. The pCaSpeR *ScrTT* construct was used to transform the *Drosophila melanogaster* strain, $y^1$ $w^{67c23.2}$ via *P*-element-mediated transformation [54].

## Ectopic expression of SCR and SCRTT and preparation of first instar cuticles

For ectopic expression of SCR, virgin female flies of the genotype, *y w; P{UAS-Scr, w+}* (Bloomington stock # 7302) were crossed with the GAL4 driver males of the genotype, *y w; P{Armadillo-Gal4, w+}* (Bloomington stock # 1560) and progeny were collected [55]. For ectopic expression of SCRTT protein from the heat-shock promoter *ScrTT* fusion gene, a heat-shock was administered at 5 hours AEL at 36.5˚C for 30 minutes. First instar cuticles were prepared as described [56] and imaged with darkfield optics on a Leica® Leitz™ DMRBE microscope.

## Ectopic expression of SCRTT from the heat-shock and UAS promoters

To ectopically express SCRTT using the GAL4-*UAS* system [55], adult virgin female flies of the genotype, *y w; P{UAS-ScrTT, w+}* were crossed with the GAL4 driver males of the genotype, *y w; P{Armadillo-Gal4, w+}* (Bloomington stock # 1560) and the progeny expressed SCRTT ubiquitously. To induce expression of SCRTT from the heat-shock promoter, *D. melanogaster* embryos at 0–16 hours AEL were collected from the apple juice plates on nylon mesh screens with an opening size of 0.3mm stretched across the bottom of a cylindrical basket 3cm in diameter and 5cm high. The embryos were heat-shocked for 30 minutes at 37.5˚C by immersion of the filter basket in a circulating water bath.

## Affinity purification of SCRTT protein from embryos

SCRTT protein was purified from embryo extracts using subcellular fractionation followed by metal affinity chromatography in denaturing conditions [57,58]. 3g of heat-shocked embryos was homogenized in 15ml of lysis buffer (15mM HEPES pH 7.6, 10mM KCl, 5mM $MgCl_2$, 2mM EDTA, 350mM sucrose, 0.032% 2-mercaptoethanol, with protease inhibitors: 0.2mM phenylmethanesulfonylfluoride (PMSF), 1.3mM benzamidine and 0.3mM Aprotinin) using a 40ml Dounce Homogenizer. The lysate was centrifuged in a Corex tube at 10,000 rpm for 15 minutes in a Sorval SS-34 rotor, and the supernatant was discarded. The white top layer of the pellet (pellet 1), leaving the dark colored debris behind, was carefully resuspended in a resuspension buffer 1 (15mM HEPES pH 7.6, 10mM KCl, 0.1mM EDTA, 350mM sucrose, 0.006% 2-mercaptoethanol, with protease inhibitors: 0.2mM phenylmethanesulfonylfluoride (PMSF), 1.3mM benzamidine and 0.3mM Aprotinin) and was centrifuged at 10,000 rpm for 15 minutes in a Sorval SS-34 rotor. The pellet (pellet 2) was resuspended in resuspension buffer 2 (15mM HEPES pH 7.6, 10mM KCl, 350mM sucrose, with protease inhibitors: 0.2mM phenylmethanesulfonylfluoride (PMSF), 1.3mM benzamidine and 0.3mM Aprotinin) and was centrifuged at 10,000 rpm for 15 minutes in a Sorval SS-34 rotor. The pellet (pellet 3) was resuspended in a nuclear lysis buffer (50mM $NaH_2PO_4$, pH 7.5, 300mM NaCl, 20mM imidazole, 1% NP-40, with protease inhibitors: 0.2mM phenylmethanesulfonylfluoride (PMSF), 1.3mM benzamidine and 0.3mM Aprotinin) and was centrifuged at 12,000 rpm for 10 minutes in a Sorval SS-34 rotor. All preceding steps were performed at 0-5ºC. The nuclear extract (NE) was mixed with solid urea to a final concentration of 8M, and the mixture was gently rocked at room temperature until the urea dissolved. The denatured nuclear extract (NE+Urea) was mixed with 250μl of Ni-NTA sepharose beads (IBA Lifesciences) that had been equilibrated with the denaturing nuclear lysis buffer, and gently rocked for 15 minutes at room temperature. The beads were packed in a column by gravity flow and the flow-through was reapplied to the column. The beads in the column were washed twice with the denaturing nuclear lysis buffer and then, washed twice with a buffer containing 50mM $NaH_2PO_4$, 300mM NaCl pH 7.5. The beads were stored at -80˚C.

## Western blot analysis

The proteins in an SDS-polyacrylamide gel were transferred onto an Immobilon®-P PVDF transfer membrane (Millipore Sigma) by electroblotting at 250mA for two hours in ice cold transfer buffer (25mM Tris, 192mM glycine and 10% methanol). The blots were blocked at room temperature for one hour in Blotto (PBT: 10% PBS and 0.1% Tween-20, and 3% skim milk). Anti-FLAG M2 monoclonal antibody (Sigma-Aldrich) at a dilution of 60,000-fold in Blotto was incubated with the blot for one hour at room temperature. After washes with PBT, horseradish peroxidase (HRP)-conjugated goat anti-mouse antibody (ThermoFisher Sci.) at a dilution of 3,000-fold in Blotto was added and incubated for one hour at room temperature. After washes with PBT, HRP was detected using SuperSignal™ West Femto Maximum Sensitivity Chemiluminescent Substrate (ThermoFisher Sci.). Digital images were recorded using a ChemiDoc™ Imaging System (Bio-Rad). For some experiments, the membrane was stripped for one hour at room temperature using Restore™ Western Blot Stripping Buffer (Thermo-Fisher Sci.) to remove the anti-FLAG antibody and was blocked at room temperature for one hour in Blotto followed by incubation with anti-β-tubulin monoclonal antibody (E7 concentrated from Developmental Studies Hybridoma Bank, University of Iowa, Iowa City, IA) at a dilution of 1,500-fold in PBT.

## Sample preparation for MS/MS

The Ni-NTA bead slurry (25μl) from the protein purification was mixed with equal volume of 2xSDS buffer (100mM Tris-HCl pH 6.8, 200mM 1,4-dithiothreitol (DTT), 4% SDS, 20% glycerol, 1% 2-mercaptoethanol, ~1 mg/ml bromophenol blue) [59] and was heated to 90˚C for 10 minutes. The 50μl sample was loaded onto a 1.5mm thick SDS-Polyacrylamide gel (11% separating and 5% stacking gel) for size separation of proteins. The gel was stained with 1 mg/ml Coomasie blue (Coomasie Brilliant Blue™ R-250 from ThermoFisher Sci.). The destained gel was stored in 5% glacial acetic acid at 4˚C. At the Functional Proteomics Facility, Western University, London, Ontario, Canada, 10 spots were picked from the gel using an Ettan® Spot-Picker™ and were in-gel digested with either trypsin (Promega), chymotrypsin (Sigma-Aldrich) or thermolysin (Promega), and the peptides subsequently lyophilized. The peptides were analyzed with a Thermo Scientific Orbitrap Elite mass spectrometer, which uses the nano LC-E-SI-Orbitrap-MS/MS technique, at the Biological Mass Spectrometry Laboratory, Western University, London, Ontario, Canada for protein identification and characterization of post-translational modifications.

## Mass spectrometry data analysis

LC-ESI-Orbitrap-MS/MS data was analyzed at the Biological Mass Spectrometry Laboratory, Western University, London, Ontario, Canada. PEAKS™ DB software versions 7, 7.5 or 8 (Bio-informatics Solutions Inc.) [60] were used to perform *de novo* sequencing and subsequent database search. PEAKS™ PTM was used to identify post-translational modifications. PEAKS™ DB uses a peptide score, which measures the quality of the peptide-spectrum match and separates the true and false identifications. Peptide score is given as $-10\log_{10}P$, where P refers to P-value. A high peptide score and a low P-value are associated with the confidence of the peptide match. The false discovery rate (FDR) was set at 1% which establishes a peptide cut-off score. A peptide must meet the cut-off score in order to be identified by PEAKS™ DB. For our analysis, a modified peptide was associated with higher confidence if the peptide score was higher than the cut-off score by 8, which corresponds to a lower P-value. Minimal ion intensity, which is the relative intensities of position-determining fragment ions in a $MS^2$ spectrum was set to 5%. Coverage is given in the analysis as percent coverage. Average Depth (AD) is the

addition of the lengths of all chemically distinct peptides identified divided by the length of the protein. Since, the proteases used often generate fragments too large or too small for analysis, the Average Depth of regions covered ($AD_{orc}$) is calculated, which is Average Depth divided by proportion of the protein covered. To distinguish between the biologically relevant PTMs and artefactual modifications that might have arisen due to chemical handling, a manual investigation of the modifications was performed.

## Phosphopeptide enrichment and C18 desalting of non-phosphopeptides

Phosphopeptide enrichment of trypsinized α-casein or SCRTT used the EasyPhos protocol employing $TiO_2$ beads [61]. The $TiO_2$ flow-through containing potential non-phosphopeptides were desalted using a C18-StageTip [62] prior to MS/MS analysis. The C18 StageTip was solvated thrice with 200μl of 80% acetonitrile, 0.2% formic acid and 19.8% water followed by centrifugation at 3,000g for 2 mins at room temperature or until no liquid remained in the tip. The C18 StageTip was equilibrated thrice with 200μl of 2% acetonitrile and 0.2% formic acid followed by centrifugation at 3,000g for 2 mins at room temperature. The $TiO_2$ flow-through was reduced in a SpeedVac to 100μl and adjusted to have a concentration of 0.2% formic acid. The sample was loaded onto the C18 StageTip and was centrifuged at 500g at room temperature until no liquid remained in the tip. The tip was then washed thrice with 200μl of aqueous buffer (2% acetonitrile, 0.2% formic acid and 97.8% water) followed by centrifugation at 500g at room temperature until no liquid remained in the tip. The peptides were eluted with 100μl of elution buffer (80% acetonitrile, 0.2% formic acid and 19.8% water) followed by centrifugation at 500g at room temperature until no liquid remained in the tip. The eluate was concentrated under vacuum using a SpeedVac to a volume of approximately 18μl and formic acid was added at a final concentration of 0.25–0.5%. The sample was analyzed by LC-MS/MS.

## Purification of SCRTT from bacteria

SCRTT expression was induced in *E. coli* BL21(DE3) transformed with pBS *ScrTT* using standard methods [51] and purified using Ni-NTA chromatography under denaturing conditions [63,64]. Protein concentration was determined by the Bradford protein assay [65].

## Bioinformatic analysis of proteomic data

*D. melanogaster* SCR protein sequence (NCBI accession number in S6 Table) was submitted to the ELM database [28] to retrieve predicted nuclear and cytoplasmic short linear motif (SLiM) sequences as HOX transcription factors interact with nuclear and cytoplasmic components [66,67]. SLiMs with any amino acids from known ordered regions of SCR were excluded from the analysis [16,68]. To determine whether a SLiM was conserved, SCR orthologous protein sequences of various protostome and deuterostome species belonging to different phyla, were retrieved from NCBI or ORCAE (only for *T. urticae*) [69] database (accession numbers in S6 Table) and a multiple sequence alignment was performed using the tools, MAFFT version 7 [70] and Clustal Omega [71]. Each predicted SLiM of SCR was manually checked for conservation across species (S10 Fig). A SLiM was considered to be conserved only if the entire predicted SLiM retrieved from the ELM database aligned perfectly in both MAFTT and Clustal Omega. If parts of a predicted SLiM and not the entire SLiM were conserved across species, that SLiM was not considered as conserved in our analysis. If a SLiM was less than four amino acids long and was conserved, it was not included in the list of conserved SLiMs (S5 Table and Fig 2). SLiMs less than five amino acids long were not considered as a conserved SLiM unless conserved beyond Diptera.

## Statistical analysis

To determine the significance of the biased distribution of serine (S), threonine (T) and tyrosine (Y) in HOX proteins to SLiMs vs. non-SLiMs and the biased phosphate distribution in SCR SLiMs vs. non-SLiMs, Fisher's Exact Test was employed [72].

# Results

## Expression of SCRTT protein

To map PTMs of SCR expressed during embryogenesis requires an initial concentrated source of protein that can be affinity purified. The CDS (expressing SCR isoform A (417 aa); FlyBase ID FBpp0081163) of *Scr* mRNA was fused in frame to the triple tag (TT) encoding C-terminal 3X FLAG, Strep II and 6X His tags [11,50] and cloned downstream of the *UAS* promoter of pUAST [11], the heat-shock promoter (*hsp*) of pCaSpeR [53] and the T7 promoter of pET-3a [51,52]. Two major systems for ectopic expression of proteins in *Drosophila* are the heat-shock inducible promoter and the GAL4-*UAS* binary system, using the strong, ubiquitous *armadillo*-GAL4 driver. The expression of SCRTT using these two ectopic expression systems were compared (Fig 1B). The heat-shock promoter resulted in higher levels of accumulation of SCRTT (Fig 1B). The fold increase of SCRTT expression of heat-shock relative to *UAS* was too great to be accurately quantified. The relative molecular mass ($M_r$) of SCRTT protein is calculated to be 49.8 [73]. However, on a Western Blot, the SCRTT protein expressed during embryogenesis ran with a higher $M_r$ of 62 (Fig 1B).

## SCRTT protein is biologically active

To assess whether SCRTT expressed from the heat-shock promoter was biologically active, the first instar larval cuticular phenotype of heat-shocked embryos expressing SCRTT was compared with the cuticles that result from expression of untagged SCR protein in all cells of the embryo using the GAL4-*UAS* system [55]. Both SCR and SCRTT ectopic expression induced ectopic T1 beards in T2 and T3, indicating that the triple tag does not interfere with the biological activity of SCR *in vivo* (Fig 1C) [11,74,75].

## Analytical workflow for affinity purification, digestion and mapping of PTMs in embryonically expressed SCRTT

For affinity purification of SCRTT protein, 3g of *D. melanogaster* embryos, collected at 0–16 hours AEL and heat-shocked for 30 minutes at 37.5˚C, were lysed. The nuclei were collected and washed, and the proteins of the nuclear extract were denatured and SCRTT was affinity purified by Ni-NTA chromatography [63,64]. The purification of SCRTT was monitored by Western Blot analysis and shows concentration of SCRTT on the Ni-NTA beads (Fig 3B). An SDS gel stained for total protein identified a band of the correct $M_r$ for SCRTT from protein extracted from the Ni-NTA beads (Fig 3D). To determine whether this purification provided the amount of SCRTT required for MS/MS, a Western Blot analysis was performed where a sample of the Ni beads containing purified embryonic SCRTT was run alongside a sample of 3500ng of SCRTT purified from bacteria, an amount which upon MS/MS analysis resulted in efficient detection of SCRTT peptides (unpublished data) (Fig 3E). The signal for the sample of the SCRTT purified from *Drosophila* embryos was 2.4-fold less than the bacterial signal suggesting the band contained about 1500ng of SCRTT. Knowing the concentration of SCRTT purified and the amount required for efficient detection of peptides, approximately 10µg of SCRTT purified from embryos was digested and processed for each sample analyzed by MS/MS.

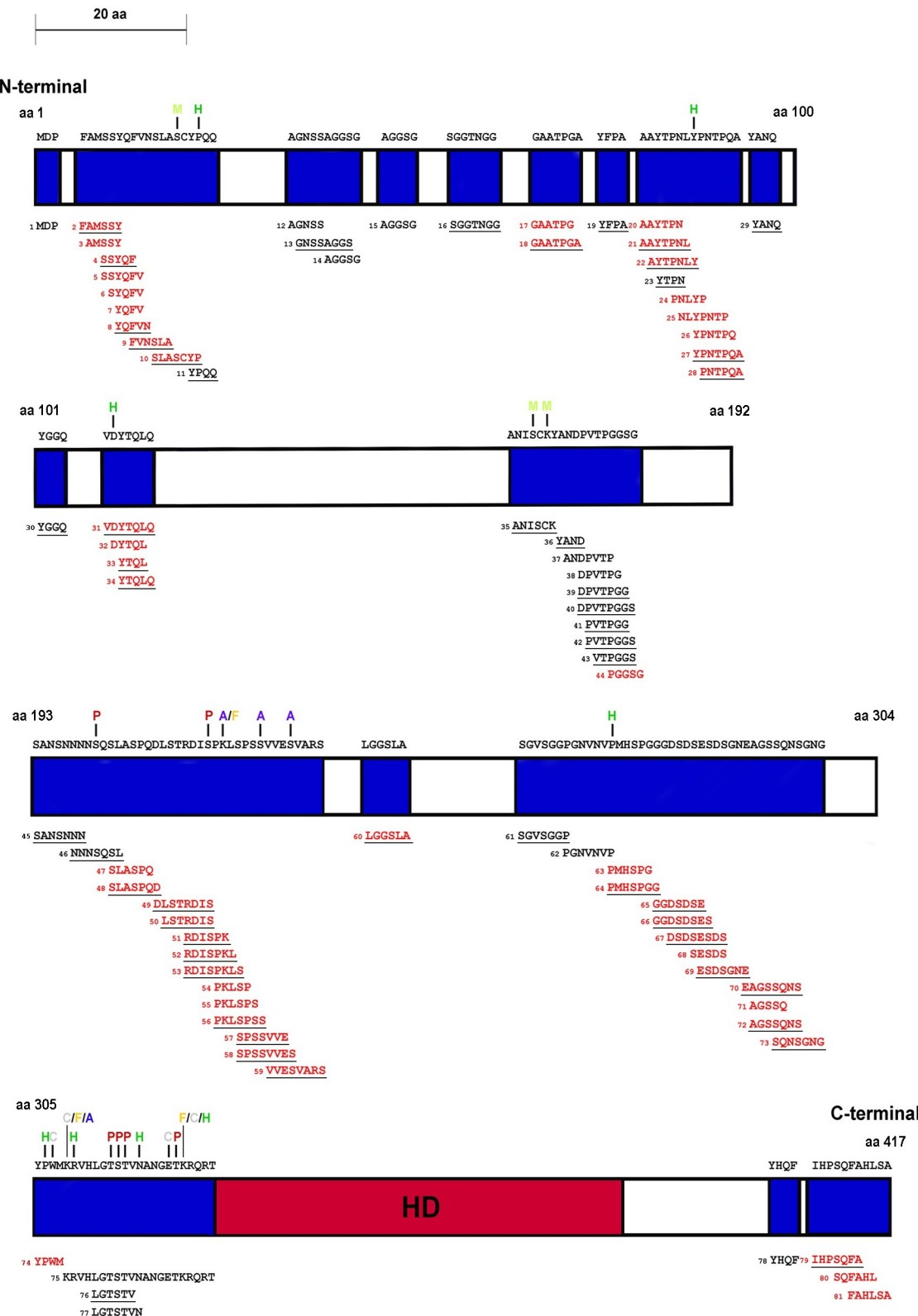

**Fig 2. Schematic of SCR showing total and conserved SLiMs.** The schematic is drawn to scale. The ordered region of the homeodomain is labeled by dark red and SLiMs are labeled blue. SLiMs that are conserved are red. The numbers beside SLiMs correspond to the SLiM data tables in the supplement (S4 and S5 Tables). The PTMs which map to SCR SLiMs are above the

modified residue and color-coded. Phosphorylation sites are indicated by P in dark red, acetylation by A in blue, formylation by F in orange, methylation by M in light green, carboxylation by C in grey and hydroxylation by H in green. SLiMs underlined are candidate phosphorylation sites.

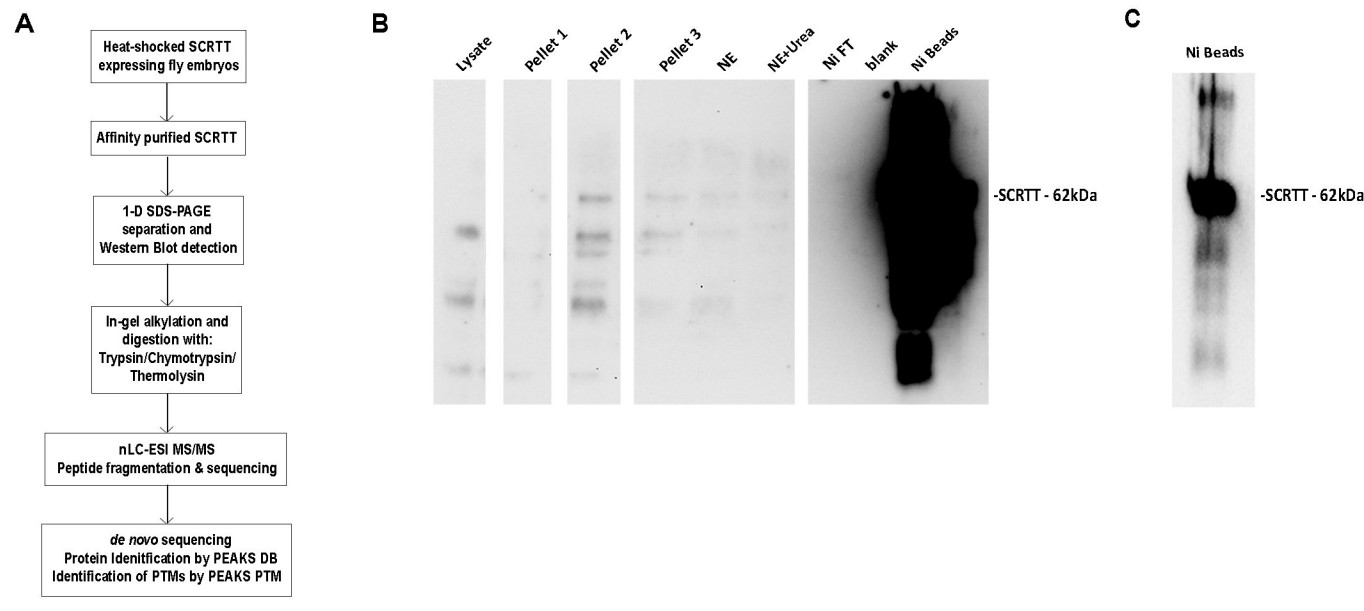

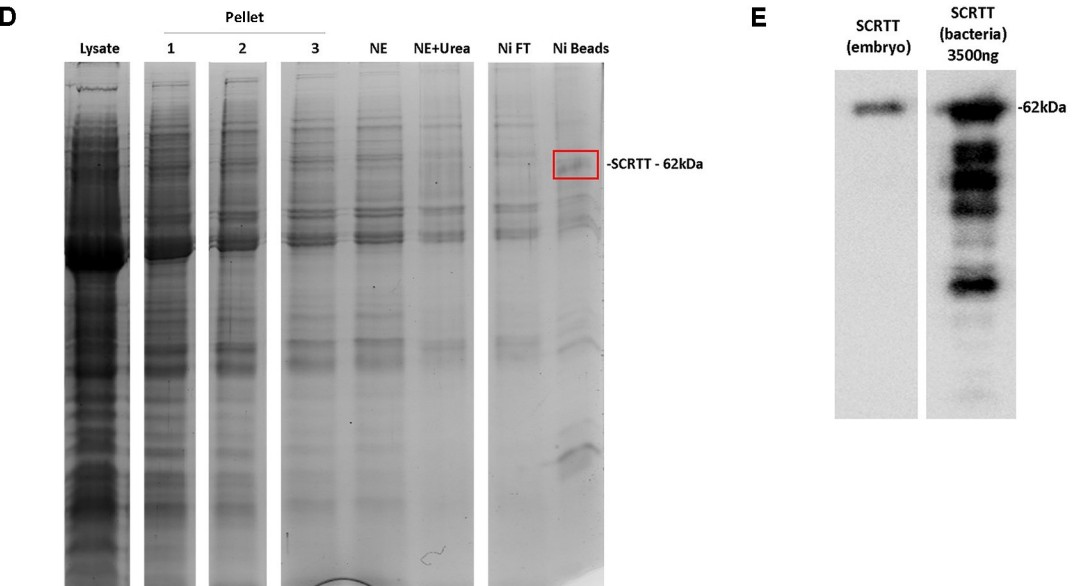

**Fig 3. Overall approach for PTM mapping of SCRTT.** (A) Analytical workflow for affinity purification, digestion and sequence mapping of PTMs in SCRTT expressed from developing embryos. (B) Overexposed Western Blot showing SCRTT at 62 kDa in the Ni beads fraction. Purification fraction is indicated at top of the lane. (C) Autoexposed Western Blot showing SCRTT at 62 kDa in the Ni beads fraction. (D) Coomasie-stained 1-D SDS-polyacrylamide gel of affinity purified SCRTT. Purification fraction is indicated at top of the lane. A band of SCRTT at 62 kDa (marked with a red box) was observed in the Ni beads fraction. (E) Comparison of *Drosophila* SCRTT vs. bacterial SCRTT to estimate the amount of protein to be analyzed by MS/MS. The Western Blot shows signals for SCRTT at 62 kDa indicated on the right. The source of SCRTT is at the top of the lane. NE Nuclear Extract, Ni Nickel and Ni FT Nickel Flow-Through. For each of the panels (B), (D) and (E), all lanes shown are from the same gel and the white space between lanes indicate the splicing out of irrelevant lanes.

Eight samples were analyzed with MS/MS of which four were digested with trypsin, two with chymotrypsin and two with thermolysin. S1 Fig shows the distribution of chemically distinct peptides over the primary sequence of SCRTT for each sample. Trypsin, chymotrypsin and thermolysin were chosen for digestion of SCRTT based on the analysis of predicted peptide generated by these enzymes [73]. The coverage and average depth of region covered ($AD_{orc}$) for each sample analyzed, and the combined coverage and $AD_{orc}$ of various combinations of samples were determined (Table 1). The final coverage was 96% of the primary sequence of SCRTT.

## Evidence for the PTMs of SCRTT

Each spectrum obtained by LC-MS/MS was interrogated by PEAKS DB search followed by identification of PTMs using PEAKS PTM algorithm. Embryonic SCRTT is post-translationally modified (S1 Fig). The modifications that were the result of a biochemical process and not a potential by-product of sample preparation were interrogated and characterized in more detail (Table 2 and S1 Fig). These 44 modifications are phosphorylation, acetylation, formylation, methylation, carboxylation or hydroxylation (S4–S9 Figs). For all modifications interrogated, six diagnostic criteria were assessed: first that the peptide had a mass shift indicative of the modification; second whether in the $MS^2$ spectra there were b and/or y ions that supported modification of a specific amino acid residue; third whether other $MS^2$ spectra were identified for peptides with a particular modification; fourth whether overlapping or differently modified peptides were identified with the same modification; fifth whether a modification was identified in multiple independent samples used for MS/MS analysis (Table 2) and sixth whether the difference of the modified peptide score and the cut-off score was greater than 8. For phosphorylation, an additional diagnostic criterion was assessed: whether b and/or y ions with a neutral loss of phosphoric acid (98 Da) were present in the $MS^2$ spectra.

## Phosphorylation of SCRTT is substoichiometric

The percentage of phosphorylated peptides is low (Table 2) indicating that phosphorylation of SCRTT may be substoichiometric. To determine whether the lack of phosphate detection was due to the instrument used for the analysis, a heavily phosphorylated protein, α-casein [76] was analyzed. 12 phosphosites were detected in α-casein (S2 Table and S2 Fig). Multiple

Table 1. MS/MS analysis–Coverage and $AD_{orc}$ for SCRTT.

| Enzyme | Sample | % Coverage | $AD_{orc}$ |
|---|---|---|---|
| Trypsin | 1 | 47 | 4.81 |
| | 2 | 66 | 10.79 |
| | 3 | 62 | 10.39 |
| | 4 | 76 | 27.29 |
| | 1+2+3+4 | 79 | 31.1[a] |
| Chymotrypsin | 5 | 80 | 8.7 |
| | 6 | 76 | 8.3 |
| | 5+6 | 82 | 12.51[a] |
| Thermolysin | 7 | 67 | 7.22 |
| | 8 | 60 | 5.92 |
| | 7+8 | 71 | 9.48[a] |
| Trypsin+Chymotrypsin+Thermolysin | - | 96 | 43.29[a] |

[a] Duplicate peptides were removed for calculation of $AD_{orc}$.

**Table 2. Summary of the evidence for PTMs in SCRTT.**

| Modification | Figure | Residue | b/y ions | Minimal ion intensity > 5% | Total no. of spectra[a] | No. of peptides modified | Percentage of peptides modified (Total no. of peptides) | No. of samples observed[b] | Peptide score—cut-off score[c] | b/y ions with a neutral loss of phosphoric acid (98 Da) |
|---|---|---|---|---|---|---|---|---|---|---|
| Phosphorylation | S3A | S185[d] | b | b | 1 | 1 | 0.66 (151) | 1 | 0.44 | b, y |
| | S3B | S201 | y | y | 1 | 1 | 0.58 (171) | 1 | 3.46 | b, y |
| | S3C | T315 | b, y | b | 1 | 1 | 0.49 (206) | 1 | 3.9 | b, y |
| | S3D | S316 | b, y | b, y | 2 | 1 | 0.49 (206) | 1 | 3.78 | b, y |
| | S3E | T317 | b, y | b, y | 3 | 2 | 0.97 (206) | 1 | 6.72 | b, y |
| | S3F | T324 | y | y | 9 | 4 | 2.31 (173) | 2 | **9.43** | b, y |
| Acetylation | S4A | K218 | y | y | 3 | 2 | 4.35 (46) | 2 | **18.87** | |
| | S4B | S223 | b, y | y | 2 | 1 | 0.92 (109) | 1 | **11.54** | |
| | S4C | S227 | b, y | - | 4 | 2 | 1.85 (108) | 1 | **14.32** | |
| | S4D | K309 | b | - | 1 | 1 | 0.93 (108) | 1 | 3 | |
| | S4E | K434 | b, y | b, y | 1 | 1 | 2 (50) | 1 | 3.2 | |
| | S4F | K439 | b, y | b, y | 2 | 1 | 1.67 (60) | 1 | 3.15 | |
| Formylation | S5A | K218 | y | y | 35 | 4 | 8.7 (46) | 4 | **26.47** | |
| | S5B | K309 | b | - | 4 | 4 | 3.7 (108) | 3 | **12.52** | |
| | S5C | K325 | b, y | y | 21 | 6 | 3.51 (171) | 3 | **17.74** | |
| | S5D | K341 | b, y | b, y | 2 | 2 | 2.99 (67) | 2 | 7.21 | |
| | S5E | K369 | b, y | b, y | 8 | 3 | 9.68 (31) | 3 | 6.03 | |
| | S5F | K434 | b, y | b, y | 2 | 1 | 2 (50) | 1 | 6.07 | |
| | S5G | K439 | b, y | b, y | 1 | 1 | 1.67 (60) | 1 | 1.63 | |
| Methylation | S6A | S19 | b, y | b, y | 6 | 2 | 9.09 (22) | 2 | **8.23** | |
| | S6B | S166 | b, y | b, y | 12 | 7 | 11.48 (61) | 2 | 2.79 | |
| | S6C | K168 | b, y | b, y | 3 | 2 | 3.33 (60) | 2 | 2.39 | |
| | S6D | T364 | b, y | y | 6 | 3 | 4.62 (65) | 2 | **13.91** | |
| Carboxylation | S7A | D108 | b, y | b, y | 6 | 2 | 1.12 (179) | 1 | **15.71** | |
| | S7B | K298 | y | - | 5 | 2 | 8.33 (24) | 2 | 3.43 | |
| | S7C | W307 | b, y | y | 16 | 6 | 4.69 (128) | 3 | **8.09** | |
| | S7D | K309 | b | - | 14 | 6 | 5.56 (108) | 3 | **13.22** | |
| | S7E | E323 | b, y | b, y | 20 | 9 | 4.89 (184) | 5 | **26.93** | |
| | S7F | K325 | b, y | b | 8 | 5 | 2.92 (171) | 2 | **10.76** | |
| | S7G | K369 | b, y | b, y | 1 | 1 | 3.23 (31) | 1 | 2.37 | |
| Hydroxylation | S8A | P22 | b, y | b, y | 1 | 1 | 4.76 (21) | 1 | 2.62 | |
| | S8B | Y87 | b, y | b, y | 1 | 1 | 2.33 (43) | 1 | 3.29 | |
| | S8C | P107 | b, y | b, y | 15 | 8 | 4.47 (179) | 7 | 7.82 | |
| | S8D | D108 | b, y | b, y | 20 | 11 | 6.15 (179) | 7 | **10.82** | |
| | S8E | D111 | b, y | b, y | 2 | 3 | 1.73 (173) | 3 | 6.05 | |
| | S8F | P269 | y | y | 6 | 5 | 12.5 (40) | 3 | **16.26** | |
| | S8G | P306 | b, y | b, y | 4 | 4 | 3.13 (128) | 2 | **19.6** | |
| | S8H | R310 | y | - | 1 | 1 | 2.94 (34) | 1 | 7.64 | |
| | S8I | N321 | b | b | 1 | 1 | 0.51 (196) | 1 | 3.66 | |
| | S8J | K325 | y | - | 1 | 1 | 0.58 (171) | 1 | **8.22** | |
| | S8K | Y334 | b, y | - | 2 | 1 | 1.69 (59) | 1 | 7.06 | |
| | S8L | R366 | y | - | 11 | 6 | 9.38 (64) | 2 | **13.6** | |
| | S8M | P392 | b, y | b, y | 1 | 1 | 9.09 (11) | 1 | 2.29 | |
| | S8N | Y398 | y | - | 1 | 1 | 10 (10) | 1 | 2.07 | |

[a] Total number of MS$^2$ spectra reporting the modification as identified by PEAKS and subsequently filtered for those with evidence from the analysis of the 8 samples (Table 1)

[b] Total number of samples the modification was observed in (Table 1)

[c] The peptide score differences greater than 8 are in bold. A peptide was associated with higher confidence if the peptide score was higher than the cut-off score by 8, which corresponds to a lower P-value.

[d] S185 phosphosite is not labelled in S1A Fig but was identified in the analysis of tryptic peptides (A2).

phosphosites map between 61–70 of α-casein, and of the 64 peptides identified for this region, 61 were phosphorylated and 24 of these peptides were phosphorylated at two amino acid residues, which is a high percentage of modified peptides indicating stoichiometric levels of phosphorylation of α-casein (S3 Fig). This suggests that phosphorylation is stable during MS/MS analysis.

TiO$_2$ beads were used to enrich for phosphopeptides from trypsinized α-casein and SCRTT samples [61]. For MS/MS analysis of α-casein, 92.3% of all peptides (84 out of 91) identified post-TiO$_2$ treatment were phosphorylated whereas without TiO$_2$ treatment, only 18.8% of all peptides (158 of 839) were phosphorylated showing that TiO$_2$ enriches for phosphopeptides (S3 Table and S2 Fig). However, only 54 phosphopeptides were common for both treatments. 104 phosphopeptides previously identified without TiO$_2$ treatment could not be identified post-TiO$_2$ treatment which indicates loss of phosphopeptides (S3 Table). Although TiO$_2$ enriches for phosphopeptides, the yield is low and no phosphopeptides were detected upon TiO$_2$ enrichment of trypsinized SCRTT.

## Are Short Linear Motifs (SLiMs) in SCR favored sites of phosphorylation?

Outside the HD, HOX proteins are highly disordered proteins [66]. SLiMs present in disordered protein regions are proposed to be preferential sites of phosphorylation [6,77,78]. To test this with SCRTT, the Eukaryotic Linear Motif (ELM) resource [28] was screened for predicted SLiMs. Of the 81 SLiMs identified, 52 (64%) are annotated as potential phosphosites. If 100% of the amino acid sequence of a predicted SLiM was conserved across species, the SLiM was considered 'conserved'. Of the 53 conserved SLiMs identified, 34 (64%) are annotated as potential phosphosites. In SCR, 63% of the primary sequence was SLiM sequence and 35% were conserved SLiM sequence (Table 3 and Fig 2). 6 out of 7 phosphosites were in a SLiM region and 1 of 7 phosphosites was in a conserved SLiM (Fig 2). There is an enrichment of phosphosites to SLiMs. Although this may suggest preferential phosphorylation of SLiMs, we also analyzed whether the amino acid residues that accept phosphates are more frequently phosphorylated in predicted SLiMs than outside of SLiMs; an additional expectation for SLiMs being preferential sites of phosphorylation. There was no significant increase in the frequency of phosphorylation of residues in SLiM versus non-SLiM regions, and a significant decrease in the frequency of phosphorylation of residues in conserved SLiMs (Table 3). In addition, 9 out of 81 SLiMs and 5 out of 53 conserved SLiMs of SCR were phosphorylated (Fig

**Table 3. SLiM analysis of SCR.**

| Percentage of SCR protein that are SLiMs | | | | | | |
|---|---|---|---|---|---|---|
| Protein (disordered) size (aa) | No. of SLiMs | SLiM size (aa) | % SLiM | No. of conserved SLiMs | Conserved SLiM size (aa) | % Conserved SLiM |
| 364 | 81 | 228 | 63 | 53 | 129 | 35 |
| Frequency of phosphorylation in SCR SLiMs vs. non-SLiMs | | | | | | |
| Percentage of S, T and Y phosphorylated | | | | *p*-value | | |
| Total SLiMs | | Conserved SLiMs | | | | |
| SLiMs | Non-SLiMs | SLiMs | Non-SLiMs | Total SLiMs | | Conserved SLiMs |
| 10 | 14 | 3 | 20 | 0.6 | | 0.04 |
| Biased distribution of S, T and Y to SCR SLiMs | | | | | | |
| Percentage of amino acid that are S, T and Y | | | | *p*-value | | |
| Total SLiMs | | Conserved SLiMs | | | | |
| SLiMs | Non-SLiMs | SLiMs | Non-SLiMs | Total SLiMs | | Conserved SLiMs |
| 26 | 5 | 28 | 13 | $2.0 \times 10^{-7}$ | | $5.7 \times 10^{-4}$ |

2 and S4 and S5 Tables) indicating that about 10% of predicted SLiMs are *bona fide* sites of SCR phosphorylation, and only 15% of SLiMs annotated as potential phosphosites are *bona fide* sites of SCR phosphorylation. This suggests that a minority of SCR predicted SLiMs were phosphorylated which also does not support the hypothesis that SLiMs are preferential sites of phosphorylation. The reason that 6 of the 7 phosphosites are in predicted SLiMs is due to the bias of S, T and Y residues to SLiMs; the percentage of S, T and Y is significantly higher in the SLiMs and conserved SLiMs than in non-SLiM portions of SCR (Table 3).

## Discussion

### PTMs of SCRTT

PTMs are one mechanism proposed for the functional specificity of HOX proteins, and mapping PTMs of HOX proteins is a first step in testing this proposal. Bottom-up MS/MS analysis of SCRTT purified from developing *D. melanogaster* embryos identified many amino acid residues that were covalently modified (Fig 4). The final analysis of PTMs did not include modifications that could be due to sample preparation; however, some of these uninterrogated modifications may be a result of a biological process that regulates SCR activity. For example, deamidation of N321, which is the second N of the NANGE motif, may have a role in DNA binding [79]. The potentially biologically relevant PTMs will be discussed in relation to conserved protein domains and sequence motifs, SCR function, distribution within predicted SLiMs and the structure of an SCR-EXD-DNA complex. Of the conserved domains/motifs of SCR important for SCR activity (Fig 1A) [6,11–16], all are post-translationally modified with the exception of the octapeptide and KMAS motifs.

### Phosphorylation of SCRTT

A clustered set of phosphorylations on the amino acid residues, T315, S316, T317 and T324 flank the NANGE motif (Fig 4). The NANGE motif is important for the suppression of ectopic proboscis formation suggesting that phosphorylation may regulate the transition of SCR activity between either determining T1 identity or determining labial identity [11]. The phosphosites map to a region of SCR not ordered in the SCR-EXD complex bound to *fkh* DNA (Fig 5). A few amino acid residues of the linker region (15 residues between YPWM motif and the HD) and the N-terminal of the HD (residues 3–9 of 60) of SCR interact stably with the minor groove of *fkh* DNA [16,81–84]. Although the amino acid residues of SCR involved in stable minor groove interactions were not found to be modified, the residues, T315, S316 and T317 that are part of the linker between the YPWM motif and HD, and T324 which is the first amino acid residue of the HD, may interact with DNA transiently. Phosphorylation adds negative charge to amino acid residues, and therefore, phosphorylation of the linker region and N-terminal arm of HD of SCR may interfere with transient minor groove interactions.

In the model for regulation of SCR activity proposed by Berry & Gehring [36], phosphorylation of the 6th and 7th amino acid residues of the HD, T329 and S330 inhibits SCR DNA binding and activity. cAMP-dependent protein kinase A phosphorylates these residues *in vitro*, and this negative regulation of SCR activity is proposed to be reversed by phosphatase PP2A-B' that removes the phosphates on these residues *in vitro*. However, loss of PP2A-B' activity had no effect on SCR activity suggesting that PP2A-B' is not involved in regulation of SCR activity [48]. Further, we have not detected phosphorylation of T329 and S330 suggesting that their phosphorylation by cAMP-dependent protein kinase A may also be an *in vitro* artifact.

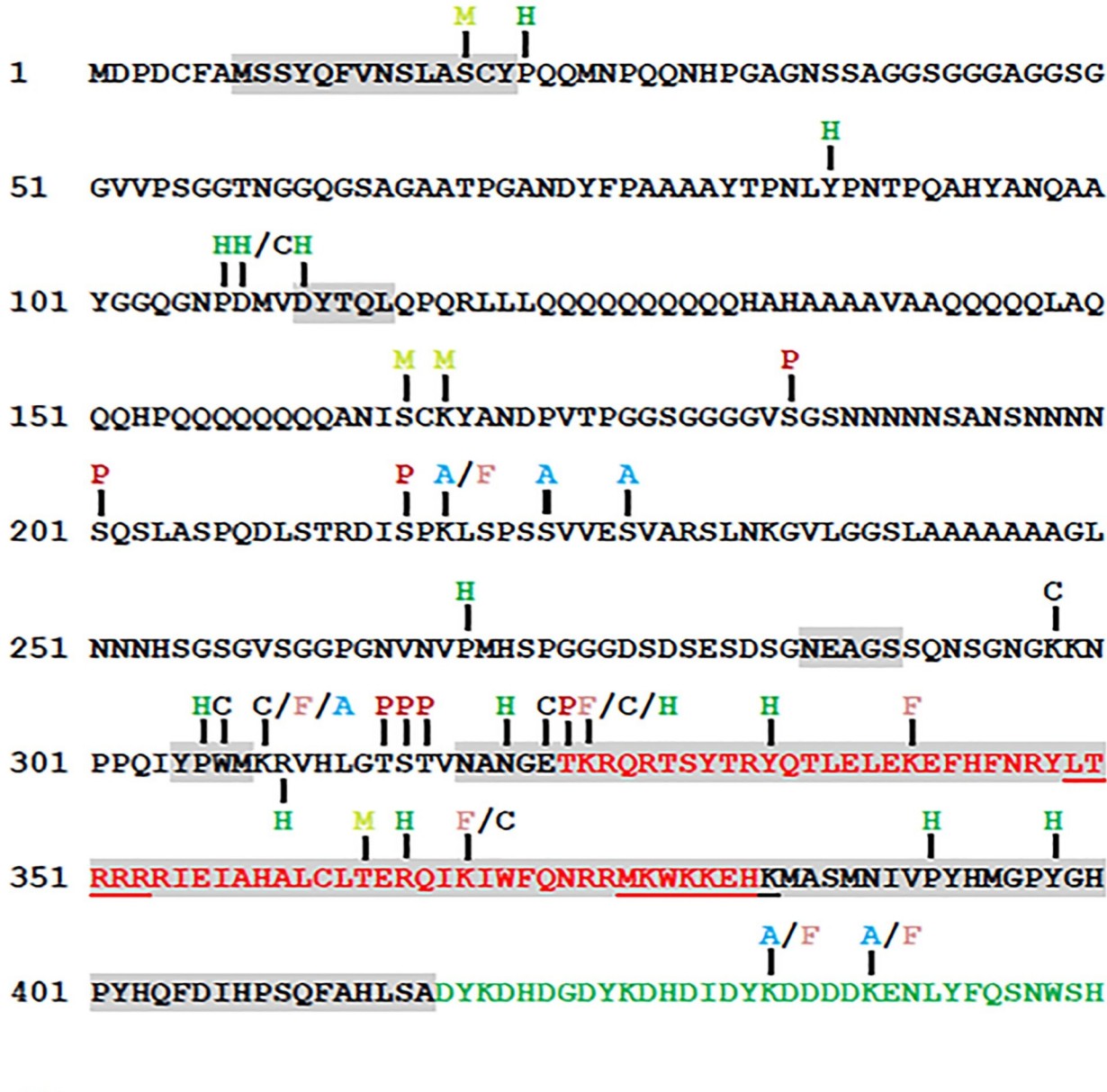

**Fig 4. A summary map of post-translational modifications in SCRTT identified by MS/MS.** Phosphorylation sites are indicated by P in dark red, acetylation by A in light blue, formylation by F in pink, methylation by M in yellowish green, carboxylation by C in black and hydroxylation by H in green. The homeodomain is highlighted in red and the triple tag is highlighted in green. The functional regions of SCR are shaded in grey (Fig 1A). The amino acids underlined were not detected by MS/MS. Phosphorylation at S216 was identified by a bulk proteomic analysis [80].

## PTMs of SCR residues found in the structure of the SCR-EXD-DNA complex

The structure of SCR-EXD bound to *fkh* regulatory DNA encompasses the evolutionarily conserved functional motifs/domains, YPWM, NANGE and HD (Fig 5). The Bilateran-specific YPWM motif of SCR and UBX binds to a hydrophobic pocket on the surface of EXD HD [16,85]. The MS/MS analysis identified hydroxylation at P306 and carboxylation at W307 of

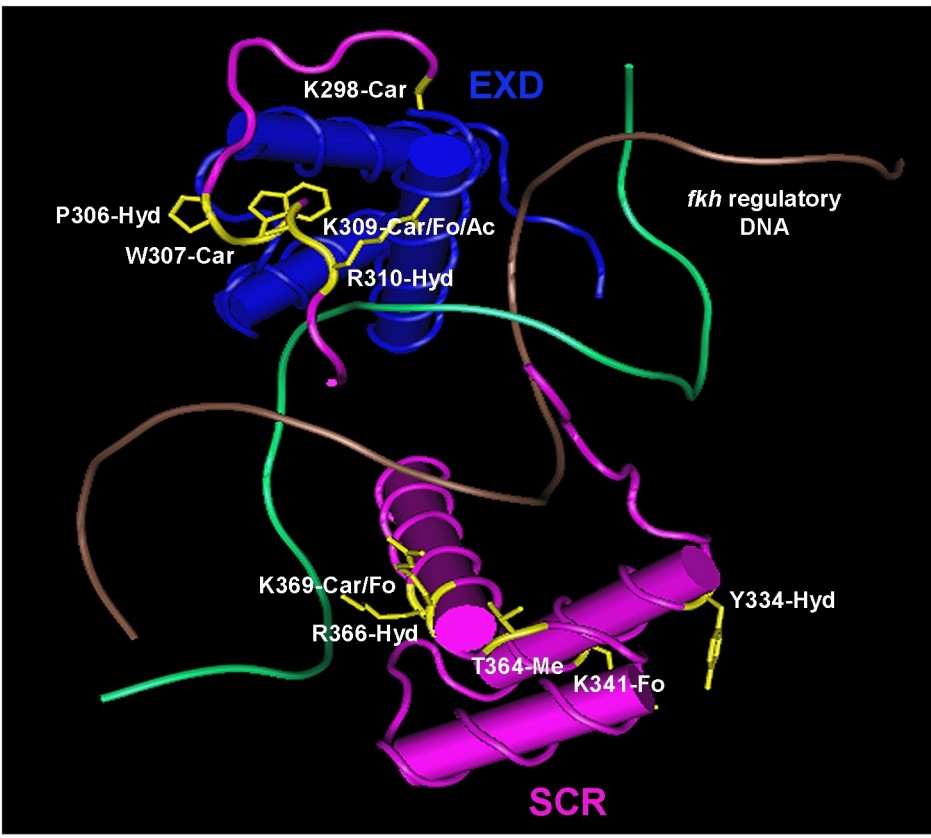

**Fig 5. The structure of SCR-EXD-DNA complex determined by crystallography.** SCR is shown in pink and EXD in blue. The two strands of *fkh* regulatory DNA is shown in brown and green. The modified amino acids of SCR along with their side chains are shown in yellow. Ac–acetylation, Car–carboxylation, Fo–formylation, Hyd–hydroxylation and Me–methylation. The structure coordinates with accession code 2R5Z (*fkh*250) were retrieved from RCSB Protein Data Bank [16]. Cn3D 4.3.1 (NCBI) was used to annotate the 3-D structure.

the YPWM motif of SCR. These modifications render the YPWM motif hydrophilic and may interfere with the binding of YPWM to the hydrophobic pocket of the EXD HD. This might be a mechanism of regulation of SCR activity as SCR-EXD interaction is essential for activating the target, *fkh* gene which is required for salivary gland development [12].

The linker region of SCR interacts with the minor groove of *fkh* DNA. Narrowing of the DNA minor groove increases the negative electrostatic potential of the groove and proteins exploit this charged state of the groove by inserting a positively charged amino acid residue, thereby, making the interaction more stable [16,83]. Although K309 and R310 residues of SCR do not directly interact with the minor groove of DNA, they render the region of the protein positively charged which aids the neighboring H312 residue in making a strong contact with the DNA minor groove [16]. Carboxylation at K309 and hydroxylation at R310 adds negative charges to this region of the protein which may be a mechanism of inhibition of the SCR-DNA interaction, thereby, regulating the functional specificity of SCR.

The highly conserved HD is a compact self-folding protein domain which interacts with the major and minor groove of DNA [3,16,86,87]. All PTMs of the SCR HD are on the solvent exposed surface. These solvent exposed residues do not interact directly with DNA, and if they have a role in regulation of SCR activity, it is unlikely due to alterations in DNA binding. The formylation of K341 and K369 may arise as a secondary modification from oxidative DNA damage when the HD is bound to DNA [88].

## Competition of acetylation and formylation observed in SCR

MS/MS analysis of SCRTT identified 4 lysine residues: K218 in the DISPK SLiM, K309 in the linker region, K434 and K439 in the triple tag that are acetylated in some peptides and formylated in others. This formylation/acetylation is found for lysine residues of core histone proteins [89].

In histones, lysine acetylation by histone acetyltransferases (HAT) and lysine deacetylation by histone deacetylases (HDAC), respectively are involved in chromatin remodeling and gene expression [90,91]. Acetylation has also been reported to modify the activity of TFs, thereby regulating the ability of the TF to bind DNA [92–94]. Besides histones, HATs and HDACs acetylate and deacetylate non-histone proteins respectively, including TFs which may be a mechanism regulating SCR activity [95–98].

Formylation of lysine residues is widespread in histones and other nuclear proteins and arise as a secondary modification due to oxidative DNA damage [88]. 5'-oxidation of DNA deoxyribose results in the formation of a highly reactive 3-formylphosphate residue which out-competes the acetylation mechanism and formylates the side-chain amino group of lysine [88]. Therefore, amino acid residues of a protein that are acetylated are also found to be formylated in many cases.

## SLiM analysis

Out of 7 sites of SCR phosphorylation found, 6 were in SLiMs suggesting that predicted SLiMs are preferential sites of phosphorylation [6]. However, there is not a significant increase in the frequency of phosphorylation of S, T and Y residues in predicted SLiMs relative to the same residues outside the SLiMs. The main reason for 6 of 7 sites of phosphorylation mapping to SLiMs is because the phosphorylatable residues, S, T and Y are concentrated in predicted SLiMs. The ELM database search yielded 81 predicted SLiMs in *D. melanogaster* SCR that span 63% of the primary sequence of the disordered region of SCR (Table 3). Filtering the predicted SLiMs for evolutionary conservation did reduce the number of SLiMs to 53. As SLiMs have a low information content [20], a database search of SCR sequence may identify false positives, which may explain why a minority of predicted SLiMs were phosphosites. However, this lack of phosphorylation may also suggest that many SLiMs are targets of other PTMs [24–28] like methylation, hydroxylation, carboxylation, acetylation and formylation. Examples of other PTMs in predicted SLiMs of SCR include a methylation at S19 and a hydroxylation at P22 of the *Drosophila*-specific SLiM, SLAS-CYP and methylation at S166 and K168 of the SLiM, ANISCK (S4 Table and Fig 2).

The analysis of conserved SLiMs suggests that they are underphosphorylated relative to the remaining SCR sequence. The SCR predicted SLiM, KRVHLGTSTVNANGETKRQRT is phosphorylated at four sites (Figs 3 and 4 and S4 Table), but this SLiM is not considered conserved in our analysis (as the entire SLiM is not conserved 100% in the *Drosophila* species chosen for our analysis, *D. pseudoobscura* and *D. virilis*) although this SLiM is conserved 100% in many other *Drosophila* species like *D. sechellia*, *D. simulans*, *D. elegans*, etc. This SLiM contains the Arthropod-specific NANGE motif which is important for suppression of ectopic proboscis formation [11]. Moreover, this SLiM maps to the linker region which interacts with the minor groove of *fkh* DNA [16,81–84]. Hence, the phosphosites may function in regulating SCR activity. If this SLiM is considered 'conserved', 5 out of 7 phosphosites would map to conserved SLiMs; however, the frequency of phosphorylation of conserved SLiMs is not greater than the frequency of phosphorylation in non-conserved SLiM regions (*p*-value = 0.7).

## Potential limitations

In order to detect PTMs by MS/MS analysis, a high concentration of the analyte protein is required which requires ectopic overexpression as endogenous protein levels are low. This

study identified PTMs on SCRTT protein expressed in all embryonic cells after administration of a heat-shock. One of the potential limitations of the ectopic approach is that the PTMs identified may not occur in cells where *Scr* is normally expressed. Another limitation of detecting PTMs on overexpressed proteins is that the heat-shock administered may affect the rate of protein modification or the amount of overexpressed protein is too high for the modification enzymes to modify completely. However, phosphoproteins expressed from heat-shock promoters are still heavily phosphorylated [38,40,41] ruling out an effect of the heat-shock on phosphorylation. An additional limitation is that PTMs are temporally and spatially regulated by the activity of modification enzymes, such as kinases and phosphatases for phosphorylation; therefore, SCR may only be post-translationally modified in a subset of developing embryonic cells resulting in detection of substoichiometric levels of modification. A final limitation is that although the total coverage of SCRTT is 96%, the depth of coverage in some regions of the protein is low and modifications in these under-represented regions may have been missed.

Although phosphorylation of proteins is common, detection of the phosphorylated amino acid residues is still a challenge. Three common explanations are used to address problems with phosphopeptide detection. Firstly, phosphopeptides are hydrophilic, and hence, they are lost during reversed-phase chromatography. Secondly, phosphopeptide ionization is selectively suppressed in the presence of unmodified peptides. Thirdly, the phosphopeptides have lower ionization or detection efficiency when compared to their unmodified moieties. There is no data to support the third argument [99]. In addition, multiply phosphorylated peptides were detected upon MS/MS analysis of a commercially purchased, pure, heavily phosphorylated, bovine α-casein protein [76] suggesting that the first two problems of phosphopeptide detection were not major issues. Therefore, the substoichiometric nature of SCRTT phosphorylation may not be a technical issue of detection but may arise during expression or purification of SCRTT. Phosphatases may remove phosphates during nuclear fractionation and nuclear lysis; however, this is unlikely during chromatography as it was performed in denaturing conditions. An attempt to enrich phosphopeptides of heavily phosphorylated α-casein using $TiO_2$ beads was successful although the yield of phosphopeptides was lower than with no enrichment. The $TiO_2$-mediated phosphopeptide enrichment proved unsuccessful for SCRTT. The MS/MS analysis of SCR protein is most likely not exhaustive and the PTMs mapped for SCR may not be the complete set that can be detected by MS/MS analysis; however, the analysis of a purified SCR protein has detected more PTMs than the bulk proteomic analyses [80,100].

Finally, our study only identified PTMs and has not assessed the functional importance directly. Two problems are associated with a functional analysis of SCR PTMs: the substoichiometric levels of modification and differential pleiotropy of *Scr* mutant alleles. Because the substoichiometric levels of modification may suggest tissue specific modifications, this whole embryo analysis is unable to suggest the tissue to assay function. Also, differential pleiotropy suggests that HOX proteins outside the HD are composed of small domains/motifs that make small tissue-specific contributions to overall HOX activity [6]. Therefore, motifs like NANGE do not have strong phenotypes when mutant making clear-cut interpretation of a genetic analysis difficult [11].

## Conclusion

This study identified sites of phosphorylation and other PTMs in a tagged HOX protein, SCRTT, extracted from developing *Drosophila melanogaster* embryos. 18 of 44 modifications map to functionally important regions of SCR. In testing the hypothesis that HOX predicted SLiMs are preferential sites of phosphorylation, we found that more phosphosites mapped to

predicted SLiMs but no support for the hypothesis that the S, T and Y residues of predicted SLiMs are more frequently phosphorylated.

## Supporting information

**S1 Fig. Coverage map of embryonic SCRTT.** Each blue line underneath the primary protein sequence of SCRTT represents a chemically distinct peptide identified by MS/MS analysis. The modifications are indicated by letters or symbols on the blue lines. The C-terminal triple tag sequence is represented by the faded region. On the right is the legend for all modifications shown in the figure. Amino acid substitutions have been excluded from the figure. (A) Analysis of tryptic peptides. A1, A2, A3 and A4 are independent MS/MS analyses. (B) Analysis of chymotryptic peptides. A1 and A2 are independent MS/MS analyses. (C) Analysis of thermolytic peptides. A1 and A2 are independent MS/MS analyses.
(PDF)

**S2 Fig. Coverage map of commercially purchased bovine α-casein.** Each blue line underneath the primary protein sequence of α-casein represents a chemically distinct tryptic peptide identified by MS/MS analysis. The modifications are indicated by letters or symbols on the blue lines. On the right is the legend for all modifications shown in the figure. Amino acid substitutions have been excluded from the figure. (A) Analysis of α-casein without $TiO_2$ treatment. (B) Analysis of α-casein with $TiO_2$ treatment.
(PDF)

**S3 Fig. Identification of phosphopeptides of a commercially purchased, pure phosphoprotein, α-casein.** The figure shows a region of 41–80 of α-casein (full protein in S2A Fig) and each blue line underneath the primary protein sequence represents a chemically distinct peptide identified by MS/MS analysis. The peptides are heavily modified, and the modifications are indicated by letters or symbols on the blue lines. On the right is the legend for all modifications shown in the figure. Amino acid substitutions have been excluded from the figure.
(PDF)

**S4 Fig. Phosphorylation of Serine 185, Serine 201, Threonine 315, Serine 316, Threonine 317 and Threonine 324 residues of SCRTT.** $MS^2$ spectra for the three phosphopeptides identified by LC-MS/MS are shown. (A) Phosphorylation of Serine 185. (B) Phosphorylation of Serine 201. The inset box shows fragment ions with *m/z* 1690 to 1790. (C) Phosphorylation of Threonine 315. (D) Phosphorylation of Serine 316. (E) Phosphorylation of Threonine 317. The inset box shows fragment ions with *m/z* 900 to 1400. (F) Phosphorylation of Threonine 324. The peptide sequence and *m/z* ratio are indicated on the top of the spectra. Positions of fragmentation are shown with vertical lines in the peptide sequence. The box on the right summarizes the evidences of phosphorylation. The relevant fragment ions and their *m/z* ratios supporting phosphorylation are labelled in the spectra.
(PDF)

**S5 Fig. Acetylation of Lysine 218, Serine 223, Serine 227, Lysine 309, Lysine 434 and Lysine 439 residues of SCRTT.** $MS^2$ spectra of the peptide identified by LC-MS/MS is shown. (A) Acetylation of Lysine 218. (B) Acetylation of Serine 223. (C) Acetylation of Serine 227. (D) Acetylation of Lysine 309. The inset box shows fragment ions with *m/z* 1150 to 1350. (E) Acetylation of Lysine 434. (F) Acetylation of Lysine 439. The inset box shows fragment ions with *m/z* 390 to 580. The peptide sequence and *m/z* ratio are indicated at the top of the spectra. Positions of fragmentation are shown with vertical lines in the peptide sequence. The box on the right summarizes the evidence confirming acetylation. The relevant fragment ions and

their *m/z* ratios supporting acetylation are labelled in the spectra.
(PDF)

**S6 Fig. Formylation of Lysine 218, 309, 325, 341, 369, 434 and 439 residues of SCRTT.** MS$^2$ spectra of the peptide identified by LC-MS/MS is shown. (A) Formylation of Lysine 218. The inset box shows fragment ions with *m/z* 300 to 700. (B) Formylation of Lysine 309. The inset box shows fragment ions with *m/z* 1100 to 1300. (C) Formylation of Lysine 325. The inset box shows fragment ions with *m/z* 700 to 1400. (D) Formylation of Lysine 341. (E) Formylation of Lysine 369. The inset box shows fragment ions with *m/z* 700 to 1100. (F) Formylation of Lysine 434. The inset box shows fragment ions with *m/z* 830 to 1010. (G) Formylation of Lysine 439. The inset box shows fragment ions with *m/z* 950 to 1200. The peptide sequence and *m/z* ratio are indicated at the top of the spectra. Positions of fragmentation are shown with vertical lines in the peptide sequence. The box on the right summarizes the evidence confirming formylation. The relevant fragment ions and their *m/z* ratios supporting formylation are labelled in the spectra.
(PDF)

**S7 Fig. Methylation of Serine 19, Serine 166, Lysine 168 and Threonine 364 residues of SCRTT.** MS$^2$ spectra of the peptide identified by LC-MS/MS is shown. (A) Methylation of Lysine 19. (B) Methylation of Serine 166. (C) Methylation of Lysine 168. (D) Methylation of Threonine 364. The inset box shows fragment ions with *m/z* 1000 to 1150. The peptide sequence and *m/z* ratio are indicated at the top of the spectra. Positions of fragmentation are shown with vertical lines in the peptide sequence. The box on the right summarizes the evidence confirming methylation. The relevant fragment ions and their *m/z* ratios supporting methylation are labelled in the spectra.
(PDF)

**S8 Fig. Carboxylation of Aspartic acid 108, Lysine 298, Tryptophan 307, Lysine 309, Glutamic acid 323, Lysine 325 and Lysine 369 residues of SCRTT.** MS$^2$ spectra of the peptide identified by LC-MS/MS is shown. (A) Carboxylation of Aspartic acid 108. (B) Carboxylation of Lysine 298. The inset box shows fragment ions with *m/z* 470 to 618. (C) Carboxylation of Tryptophan 307. The inset box shows fragment ions with *m/z* 780 to 1030. (D) Carboxylation of Lysine 309. The inset box shows fragment ions with *m/z* 990 to 1300. (E) Carboxylation of Glutamic acid 323. (F) Carboxylation of Lysine 325. (G) Carboxylation of Lysine 369. The inset box shows fragment ions with *m/z* 840 to 1020. The peptide sequence and *m/z* ratio are indicated at the top of the spectra. Positions of fragmentation are shown with vertical lines in the peptide sequence. The box on the right summarizes the evidence confirming carboxylation. The relevant fragment ions and their *m/z* ratios supporting carboxylation are labelled in the spectra.
(PDF)

**S9 Fig. Hydroxylation of Proline 22, Tyrosine 87, Proline 107, Aspartic Acid 108, Aspartic acid 111, Proline 269, Proline 306, Arginine 310, Asparagine 321, Lysine 325, Tyrosine 334, Arginine 366, Proline 392 and Tyrosine 398 residues of SCRTT.** MS$^2$ spectra of the peptide identified by LC-MS/MS is shown. (A) Hydroxylation of Proline 22. The inset box shows fragment ions with *m/z* 835 to 920. (B) Hydroxylation of Tyrosine 87. (C) Hydroxylation of Proline 107. The inset box shows fragment ions with *m/z* 670 to 1000. (D) Hydroxylation of Aspartic acid 108. The inset box shows fragment ions with *m/z* 620 to 900. (E) Hydroxylation of Aspartic acid 111. The inset box shows fragment ions with *m/z* 420 to 600. (F) Hydroxylation of Proline 269. The inset box shows fragment ions with *m/z* 420 to 940. (G) Hydroxylation of Proline 306. (H) Hydroxylation of R310. (I) Hydroxylation of N321. (J) Hydroxylation of K325. (K) Hydroxylation of Y334. The inset box shows fragment ions with *m/z* 788 to 1040. (L) Hydroxylation of R366. (M) Hydroxylation of P392. The inset box shows

fragment ions with *m/z* 750 to 890. (N) Hydroxylation of Y398. The inset box shows fragment ions with *m/z* 945 to 1010. The peptide sequence and *m/z* ratio are indicated at the top of the spectra. Positions of fragmentation are shown with vertical lines in the peptide sequence. The box on the right summarizes the evidence confirming hydroxylation. The relevant fragment ions and their *m/z* ratios supporting hydroxylation are labelled in the spectra.
(PDF)

**S10 Fig. Alignment of SLiMs of animal SCR homologs.** SLiMs of *D. melanogaster* SCR which are conserved across various taxonomic groups were aligned using multiple sequence alignment tools, MAFFT v. 7 and Clustal Omega. The phylogenetic tree on the left is not drawn to scale and it merely depicts the relationship among the organisms and not evolutionary time of divergence. The block diagram below the aligned sequences shows the HD and four SLiMs conserved beyond Diptera.
(PDF)

**S1 Raw Images. Original blot and gel images.**
(PDF)

**S1 Table. Post-translational modifications of embryonic SCRTT identified by LC-MS/MS.**
(PDF)

**S2 Table. Phosphosites in bovine α-casein identified by MS/MS.**
(PDF)

**S3 Table. Phosphopeptides of α-casein identified by MS/MS (TiO2-enriched vs. non-TiO2).**
(PDF)

**S4 Table. SLiMs in *D. melanogaster* SCR.**
(PDF)

**S5 Table. Conserved SLiMs in *D. melanogaster* SCR.**
(PDF)

**S6 Table. Accession numbers of animal SCR protein homolog sequences (retrieved from NCBI).**
(PDF)

## Acknowledgments

We thank Paula Pittock from the Biological Mass Spectrometry Laboratory, The University of Western Ontario, London, Ontario, Canada for analyzing the peptide samples by LC-ESI MS/MS and assisting us with the PTM characterization using the PEAKS™ DB software, Victoria Clarke and Kristina Jurcic from the Functional Proteomics Facility, The University of Western Ontario, London, Ontario, Canada for their assistance with the Ettan® SpotPicker™ instrument and in-gel digestion of our samples, Lovesha Sivanantharajah for providing us with the pBS *ScrTT* construct, Ben Rubin for his suggestions on the statistical analysis, Stuart Cameron and Gurjit Randhawa for assistance with the project. We extend our gratitude to René Rezsohazy for valuable suggestions on the manuscript. We thank the two reviewers for their helpful suggestions.

## Author Contributions

**Conceptualization:** Anirban Banerjee, Anthony Percival-Smith.

**Data curation:** Anirban Banerjee.

**Formal analysis:** Anirban Banerjee.

**Funding acquisition:** Anthony Percival-Smith.

**Investigation:** Anirban Banerjee, Anthony Percival-Smith.

**Methodology:** Anirban Banerjee, Anthony Percival-Smith.

**Project administration:** Anthony Percival-Smith.

**Resources:** Anirban Banerjee.

**Software:** Anirban Banerjee.

**Supervision:** Anthony Percival-Smith.

**Validation:** Anirban Banerjee.

**Visualization:** Anirban Banerjee.

**Writing – original draft:** Anirban Banerjee.

**Writing – review & editing:** Anirban Banerjee, Anthony Percival-Smith.

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
