## [Decision Letter · Decision Letter 0]

15 Oct 2019

PONE-D-19-26238

Post-translational modifications of Drosophila melanogaster HOX protein, Sex combs reduced

PLOS ONE

Dear Banerjee,

Thank you for submitting your manuscript to PLOS ONE. After careful consideration, we feel that it has merit but does not fully meet PLOS ONE’s publication criteria as it currently stands. Therefore, we invite you to submit a revised version of the manuscript that addresses the points raised during the review process.

The reviewers have identified a number of major and minor concerns. It is important to fully address them in your revised manuscript. These include, but are not limited to, providing evidence about the functional significance of the discovered PTMs, providing more details and further clarifying the results related to SLiMs, among other issues raised by the reviewers.

We would appreciate receiving your revised manuscript by Nov 29 2019 11:59PM. To enhance the reproducibility of your results, we recommend that if applicable you deposit your laboratory protocols in protocols.io, where a protocol can be assigned its own identifier (DOI) such that it can be cited independently in the future. For instructions see: http://journals.plos.org/plosone/s/submission-guidelines#loc-laboratory-protocols

We look forward to receiving your revised manuscript.

Kind regards,

Jinsong Zhang

Academic Editor

PLOS ONE

Journal Requirements:

Reviewers' comments:

Reviewer's Responses to Questions

**Comments to the Author**

1. Is the manuscript technically sound, and do the data support the conclusions?

Reviewer #1: Yes

Reviewer #2: Yes

2. Has the statistical analysis been performed appropriately and rigorously? 

Reviewer #1: I Don't Know

Reviewer #2: N/A

3. Have the authors made all data underlying the findings in their manuscript fully available?

Reviewer #1: Yes

Reviewer #2: Yes

4. Is the manuscript presented in an intelligible fashion and written in standard English?

Reviewer #1: Yes

Reviewer #2: Yes

5. Review Comments to the Author

Reviewer #1: The MS by Banerjee et al. reports on the extensive characterization of postranslational modifications (PTMs) targeting the Drosophila Hox protein Sex combs reduced (Scr). This is to my knowledge the first extensive description of Hox associated PTMs, providing qualitative and quantitative insights of how PTMs can impact Hox proteins. Consistent with role of PTMs affecting Hox protein function, a large proportion of the identified PTMs targets regions of the Scr protein previously known to be functionally important. The MS however do not experimentally address the functional relevance of any of the PTMs identified.

The MS is overall well written, the material and methods are extensive, and Fig. 7 and 8 nicely summarizes the content of the MS. Some Figures with low info content may be combined (Fig.1 to 3).

Specific comments:

- The authors address whether phosphorylation sites preferentially map to SLiM motifs. It is not clear why this specific question is raised, and also why it is addressed only for phosphorylation.

- Fig. 1 highlights FUNCTIONAL Scr protein domains. The bases for that claim for each motif should be provided. Also it may be useful to complement that figure to show to what extent these domains are conserved over short and/or long evolutionary distances.

- Fig. 2 shows a single Scr band, raising the issue of whether/how much of these PTMs likely “coexist”. May be a more resolutive gel and less exposed picture would help appreciating the point.

- Quality of the Figure may be improved for a few panels (in Fig. 3, larvae are apparently not at equivalent stages, and one panel of Fig. 4B is strongly over exposed).

- Discussion, p32, the statement “that the modifications map to functionally important region” is not supported. This point is accurately described in the abstract.

- The authors do not discuss one important limitation that relates the “ectopic” approach. Scr is only expressed very locally in the Drosophila embryo. Following heat shock it is expressed at high levels in all embryonic cells, which may reveal PTMs that do not occur in cells “normally” expressing Scr.

Reviewer #2: This manuscript reports results of a study of the post-translational modifications of the Drosophila Hox protein SCR. The study design was to overexpress a tagged version of Scr in embryos, extract it and analyze modifications by Tandem Mass Spectrometry. The study identified multiple protein modifications including phosphorylation, acetylation, formylation, carboxylation and hydroxulation for a total of 44 modifications. Many of these map to regions of the protein previously shown to be important for function. The study then did bioinformatics analysis to identify SLiMs and found that 66% of SCR is comprised of these, of which >30% are evolutionarily conserved.

The report of PTMs for SCR is of interest and worthy of publication. This analysis appears to have been carefully done and provides worthwhile information for the study of Hox protein function in general. I have two major reservations:

1. The PTMs were analyzed on SCR protein that was expressed throughout the embryo , at high levels and over a very long time period. While this study design is understandable, in that large amounts of protein are needed, it raises the problem that expression outside of the native SCR domain may result in PTMs that do not occur in the cells in which SCR is endogenously expressed. Therefore, the functional significance of the different modifications is unclear. I would like to see at least some functional testing of residues – perhaps at least the phosphorylated residues predicted to be in regions important for function – in order to be able to evaluate the significance of the PTMs identified.

2. The finding that most of SCR is covered by SLiMs is very surprising to me and I am, quite frankly, skeptical about this result and its potential importance. At a minimum, this needs a lot more explanation. How do you define a SLiM? What are your cut-offs? What are you comparing? Same questions for the conservation. What is compared? How broadly conserved are these? Across drosophilid SCRs? Other species? I would add main text figures on this, and show the sequence alignments with sequences (not just blocks). The Tables in the Supplement are fine but are basically just lists. The main text needs to summarize this information in more than just the schematic shown in the main Figure. I really wonder what the significance of all these is? Would you find this many SLiMs in any protein analyzed? This needs to be tested in order to understand whether this finding is significant.

Other specific comments:

Line 257: Indicate C-terminal or N-terminal fusion; replace “behind” with “downstream of”

Line 262: This is an enormous difference. It might be worth stating here which GAL4 was used (even though it's in the methods) and perhaps commenting on whether that is considered a "strong" GAL4 line. Just to point to how generalizable this is - that it is so much stronger (which is something many of us assume but I have not noticed it directly compared like this before to a ubiquitous GAL4 driver)

Line 287: It would be helpful if the WT control were imaged in a way that shows the T1 beard. Also, it is now pretty standard to use a UAS-gfp as control rather than a WT animal

Line 291: Perhaps worth a mention here or in the methods how you managed the heat shock with 3 grams of embryos - size of tools used to collect, size of mesh - just a few words to point out that this is a larger mass of embryos than your standard collection. Also, please include stage and length of heat shock, here even though it is in the methods

Line 300: This is confusing if you are saying that the purified band had less than you need for the MS. You just need more sentences here to explain. It is also notable that a lot of background bands came through the whole purification. Obviously you got the right product in the end but I don't know if you want to comment on that point when discussing Fig. 4. You really cannot see the SCR band above background levels in the Coomassie stained gel.

Fig. 5: I would make this a supplementary figure. It's kind of distracting to the main story line to see a figure with sequence and then that sequence isn't SCR.

6. PLOS authors have the option to publish the peer review history of their article (what does this mean?). If published, this will include your full peer review and any attached files.

Reviewer #1: No

Reviewer #2: Yes: Leslie Pick

---

## [Author Response · Author response to Decision Letter 0]

23 Nov 2019

The file named 'Response to Reviewers' contains responses to all points raised by the reviewers. Thank you.

---

## [Decision Letter · Decision Letter 1]

10 Dec 2019

PONE-D-19-26238R1

Post-translational modifications of Drosophila melanogaster HOX protein, Sex combs reduced

PLOS ONE

Dear Banerjee,

Thank you for submitting your manuscript to PLOS ONE. After careful consideration, we feel that it has merit but does not fully meet PLOS ONE’s publication criteria as it currently stands. Therefore, we invite you to submit a revised version of the manuscript that addresses the points raised during the review process.

Although Reviewer 1 has indicated that your revised manuscript has addressed the previous concerns, Reviewer 2 has remaining concerns that need to be further addressed, which include clarification of the species related to the "conserved" SLiMs, and some validation of the significance of the predicted SLiMs from the database used in the manuscript.

We would appreciate receiving your revised manuscript by Jan 24 2020 11:59PM. To enhance the reproducibility of your results, we recommend that if applicable you deposit your laboratory protocols in protocols.io, where a protocol can be assigned its own identifier (DOI) such that it can be cited independently in the future. For instructions see: http://journals.plos.org/plosone/s/submission-guidelines#loc-laboratory-protocols

We look forward to receiving your revised manuscript.

Kind regards,

Jinsong Zhang

Academic Editor

PLOS ONE

Reviewers' comments:

Reviewer's Responses to Questions

**Comments to the Author**

1. If the authors have adequately addressed your comments raised in a previous round of review and you feel that this manuscript is now acceptable for publication, you may indicate that here to bypass the “Comments to the Author” section, enter your conflict of interest statement in the “Confidential to Editor” section, and submit your "Accept" recommendation.

Reviewer #1: All comments have been addressed

Reviewer #2: (No Response)

2. Is the manuscript technically sound, and do the data support the conclusions?

Reviewer #1: Yes

Reviewer #2: Yes

3. Has the statistical analysis been performed appropriately and rigorously? 

Reviewer #1: I Don't Know

Reviewer #2: I Don't Know

4. Have the authors made all data underlying the findings in their manuscript fully available?

Reviewer #1: Yes

Reviewer #2: Yes

5. Is the manuscript presented in an intelligible fashion and written in standard English?

Reviewer #1: Yes

Reviewer #2: Yes

6. Review Comments to the Author

Reviewer #1: The authors have properly addressed the issue raised in my first round of review and modified accordingly the manuscript.

Reviewer #2: This revised manuscript is much improved and many of my suggestions have been addressed. I can agree with the argument that this study is worthy of publication before functional testing because of the complications that those experiments present, as discussed by the authors both in the rebuttal and in the text. The one additional major point I would like to see addressed though is the finding that so much of the SCR protein is comprised of SLiMs. First, the authors should make clear in the text and figure that “conserved” in this case refers to Drosophila species. This is quite different from an earlier figure where conserved included all arthropods - hundreds of millions of years of difference in divergence. Second, I ask that the authors experimentally (bioinformatically) address the question I raised previously: Would you find this many SLiMs in any protein analyzed? This needs to be tested in order to understand whether this finding is significant, especially since experimental testing of individual sequences is not realistic in this system. Are there examples of proteins analyzed in this way for which the function of the predicted SLiMs has been experimentally tested or verified? Basically, the conclusion now is based on analysis in one database. I personally am not familiar with the stringency of this database but I am still very skeptical about the finding. I would be happy to be convinced of its validity – it would help our own research enormously – but I need more evidence. Most of the other points were addressed adequately although I did not some unusual grammar and punctuation in some of the changes – it needs a quick proofreading before resubmission.

7. PLOS authors have the option to publish the peer review history of their article (what does this mean?). If published, this will include your full peer review and any attached files.

Reviewer #1: No

Reviewer #2: Yes: Leslie Pick

---

## [Author Response · Author response to Decision Letter 1]

15 Dec 2019

Dear Editor,

We thank Reviewer #1 for accepting the revisions to the manuscript and Dr. Leslie Pick (Reviewer #2) for her helpful comments on the manuscript, and we are sure that the advice taken will greatly improve the manuscript. Dr. Pick’s comments are in italics and our responses are in roman.

The page and line numbers mentioned below correspond to the manuscript without track changes with file name ‘Manuscript’.

Reviewer #2 Dr. Leslie Pick

1. The one additional major point I would like to see addressed though is the finding that so much of the SCR protein is comprised of SLiMs. 

We have addressed this point in the Discussion section (lines 533-539). 

“The ELM database search yielded 81 predicted SLiMs in D. melanogaster SCR that span 63% of the primary sequence of the disordered region of SCR (Table 3). Filtering the predicted SLiMs for evolutionary conservation did reduce the number of SLiMs to 53. As SLiMs have a low information content [20], a database search of SCR sequence may identify false positives, which may explain why a minority of predicted SLiMs were phosphosites. However, this lack of phosphorylation may also suggest that many SLiMs are targets of other PTMs [24-28] like methylation, hydroxylation, carboxylation, acetylation and formylation.”

2. First, the authors should make clear in the text and figure that “conserved” in this case refers to Drosophila species. This is quite different from an earlier figure where conserved included all arthropods - hundreds of millions of years of difference in divergence. 

SCR SLiMs predicted by the ELM database (S4 Table) were manually checked for conservation across species which included protostomes and deuterostomes (lines 263-271). 39 out of 53 “conserved” SLiMs are conserved among Drosophilids. However, 14 “conserved” SLiMs are conserved beyond Drosophila like VDYTQLQ (SLiM #31) which is conserved across Diptera, YTQL (SLiM #33) which is conserved across Insecta, SSYQF (SLiM #4) which is conserved across Arthropoda, YPWM (SLiM# 74) which is conserved across Bilateria, etc. The extent of conservation of “conserved” SLiMs is documented in S5 Table.

3. Second, I ask that the authors experimentally (bioinformatically) address the question I raised previously: Would you find this many SLiMs in any protein analyzed? This needs to be tested in order to understand whether this finding is significant, especially since experimental testing of individual sequences is not realistic in this system. 

Yes, the low information content of SLiMs does lead to a large number of predicted SLiMs in any protein. We randomly chose 8 D. melanogaster nuclear proteins and submitted their amino acid sequences to the ELM database to find predicted SLiMs in those proteins. 9-75% of the total protein was SLiMs with an average of 42%. For HOX proteins in D. melanogaster, 46-64% were SLiM sequences with an average of 57% (refer to tables below).

D. melanogaster nuclear protein UniProt ID Protein size (aa) Predicted SLiM size (aa) % SLiMs Mean % SLiMs

Cyclin-dependent kinase 8 Q9VT57 454 42 9 

42 ± 21.9

Helicase Q9VRI0 560 185 33 

Nucleoporin seh1 Q7K2X8 354 98 28 

AT07420p Q7JWU9 249 108 43 

RNA-binding protein 8A Q9V535 165 86 52 

Histone RNA hairpin-binding protein Q9VAN6 276 206 75 

Cyclin-C P25008 267 65 24 

Histone chaperone asf1 Q9V464 218 160 73 

D. melanogaster HOX protein

 Protein (disordered) size (aa) SLiM

size (aa) % SLiM Mean % SLiMs

LAB 576 366 64 

57 ± 5.5

PB 724 461 64 

DFD 533 287 54 

SCR 364 228 63 

ANTP 325 175 54 

UBX 336 156 46 

ABD-A 277 156 56 

ABD-B(m) 440 251 57 

ABD-B(r) 217 125 58 

However, bioinformatic analysis has shown that human disease mutations in intrinsically disordered regions do map frequently to validated and predicted SLiMs indicating that SLiMs are functional. The function of SLiMs has been tested in many proteins both viral and host. We have included this evidence in the introduction when discussing SLiMs (lines 63-76).

“SLiMs or Eukaryotic Linear Motifs (ELMs) are short stretches of protein sequence (typically 3-10 amino acids long containing 2-3 specificity-determining residues) present in regions of intrinsic disorder [20-23]. SLiMs function as sites of protein-protein interaction, PTMs and cell compartment targeting signals [20-28]. Experimentally validated SLiMs have been curated into the ELM database and the database is used to predict SLiMs [28]. Although SLiMs have a low information content and are predicted to occur frequently in the proteome, the 22% of human disease mutations that occur in intrinsically disordered regions of human proteins map frequently to functionally important residues of validated SLiMs and also map frequently to predicted SLiMs suggesting that predicted SLiM sequences are correlated with function [29]. Viral proteins have SLiMs that mimic the SLiMs in host proteins which enables the virus to evade host immune functions [30,31]. Phosphorylation of the PDZ-binding motif (PBM), which is a SLiM, in the disordered C-terminal tail of ribosomal S6 kinase 1 (RSK1) protein is important for epidermal growth factor regulation [32]. The annotation of SLiMs in the ELM database suggest that a significant proportion are targets of phosphorylation.”

4. Are there examples of proteins analyzed in this way for which the function of the predicted SLiMs has been experimentally tested or verified? 

We give one of many examples in the introduction. See above. Also, the ELM database is a curated collection of validated SLiMs which is searched to find predicted SLiMs. We also looked for conservation of the SLiMs as an additional criterium for the potential of function.

5. Basically, the conclusion now is based on analysis in one database. I personally am not familiar with the stringency of this database but I am still very skeptical about the finding. I would be happy to be convinced of its validity – it would help our own research enormously – but I need more evidence. 

We understand Leslie Pick’s concern, which is why we assessed whether predicted SLiMs were preferential sites of phosphorylation and they are not, suggesting some skepticism about the use of the SLiM database to predict phosphosites beyond just having the three amino acids that are phosphorylated. However, as we now point out in the manuscript, human disease mutations map frequently to SLiMs predicted from the ELM database, suggesting that this database has a use in identifying potential functional protein elements. Short functional sequences are difficult to predict and work with and tools like the ELM database need to be developed and tested for their usefulness. 

6. Most of the other points were addressed adequately although I did not some unusual grammar and punctuation in some of the changes – it needs a quick proofreading before resubmission.

The revised manuscript has been thoroughly checked for grammar and punctuation.

---

## [Editor Report · Decision Letter 2]

26 Dec 2019

Post-translational modifications of Drosophila melanogaster HOX protein, Sex combs reduced

PONE-D-19-26238R2

Dear Dr. Banerjee,

We are pleased to inform you that your manuscript has been judged scientifically suitable for publication and will be formally accepted for publication once it complies with all outstanding technical requirements.

With kind regards,

Jinsong Zhang

Academic Editor

PLOS ONE
---

## [Editor Report · Acceptance letter]

30 Dec 2019

PONE-D-19-26238R2 

Post-translational modifications of *Drosophila melanogaster* HOX protein, Sex combs reduced 

Dear Dr. Banerjee:

I am pleased to inform you that your manuscript has been deemed suitable for publication in PLOS ONE. Congratulations! Your manuscript is now with our production department. 

With kind regards,

on behalf of

Dr. Jinsong Zhang 

Academic Editor

PLOS ONE